# Lignocellulolytic Biocatalysts: The Main Players Involved in Multiple Biotechnological Processes for Biomass Valorization

**DOI:** 10.3390/microorganisms11010162

**Published:** 2023-01-08

**Authors:** Ana Laura Totti Benatti, Maria de Lourdes Teixeira de Moraes Polizeli

**Affiliations:** Department of Biology, Faculty of Philosophy, Science and Letters of Ribeirão Preto, University of São Paulo, São Paulo 14040-901, Brazil

**Keywords:** microorganisms, lignocellulolytic enzymes

## Abstract

Human population growth, industrialization, and globalization have caused several pressures on the planet’s natural resources, culminating in the severe climate and environmental crisis which we are facing. Aiming to remedy and mitigate the impact of human activities on the environment, the use of lignocellulolytic enzymes for biofuel production, food, bioremediation, and other various industries, is presented as a more sustainable alternative. These enzymes are characterized as a group of enzymes capable of breaking down lignocellulosic biomass into its different monomer units, making it accessible for bioconversion into various products and applications in the most diverse industries. Among all the organisms that produce lignocellulolytic enzymes, microorganisms are seen as the primary sources for obtaining them. Therefore, this review proposes to discuss the fundamental aspects of the enzymes forming lignocellulolytic systems and the main microorganisms used to obtain them. In addition, different possible industrial applications for these enzymes will be discussed, as well as information about their production modes and considerations about recent advances and future perspectives in research in pursuit of expanding lignocellulolytic enzyme uses at an industrial scale.

## 1. Introduction

Biodiversity is a broad concept that covers all the forms and combinations of life variations in all biological organization levels. All the forms of life existent on Earth, including plants, animals, microorganisms, the genes they contain, and the ecosystems they form, are part of the planet’s biodiversity [1]. 

Estimates made by researchers show that there are around 9 million species on the planet nowadays [2]. However, for the last 200 years, population and socio-economic activities have significantly grown, causing various pressures on the planet’s natural resources [3,4]. Many researchers claim we face a 6ª Earth-mass extinction [5,6]. Studies show that current extinction rates are from 1000 to 10,000 times higher than previous ones registered by the fossil register due to natural causes. If extinction rates and species description continue, both progressing in these opposites ways, it will take many species to extinction even before we know them [2].

To reduce the anthropic impact on the environment, many areas have been studying new techniques to be applied. In that regard, biotechnology emerges as a tool capable of increasing food security and productivity by occupying less area. In addition, it reduces greenhouse gas emissions through more sustainable technologies for energy and food production, biofuel production, and bioremediation, among others [7,8,9,10,11,12]. This way, biotechnology is considered a meaningful way to revert the current environmental crisis scenario [13,14]. This microorganism’s metabolic capacity is so great that numerous are products, industrials process, and technologies which derive their metabolites. These fields are included food, agroindustry, chemical industry, biofuels, medicines, and other materials [14,15,16]. Among all products from microbial metabolism, enzymes have significant importance [17]. 

## 2. Microorganisms as Enzymes Sources

Microorganisms are the oldest and most widespread living beings on the planet. About 3.8 to 3.9 billion years ago, they already inhabited the Earth, way before the appearance of plants and animals. Despite being the smallest forms of life, they constitute the most significant part of the biomass existent on the planet [18]. Most recent studies estimate that, among the prokaryotes (including archaea and bacteria), there are around 0.8 to 1.6 million species [19]. Among eukaryotes, fungi represent one of the most significant e most diverse kingdoms, with estimates of approximately 6 million surviving species [20]. Most of these species are found in soil and underground ocean environments, although some of them can inhabit extreme environments such as hot and hydrothermal springs, polar ice caps, as well as hypersaline and extreme pH environments [21,22,23]

Since antiquity, humankind has benefited from microorganisms’ use in the most diverse process, such as bread fermentation, preparation of alcoholic beverages, cheese, and fermented milk preparation, and growing crops for food [24,25,26]. However, despite having many ancient applications, the biotechnology field from microorganisms is one of the most recent but also the one with the most considerable growth among industries [27].

Currently, microorganisms are the primary source of obtaining enzymes with industrial applications. The use of enzymes in most different industries has gained more and more relevance and importance. Since the population grows at elevated levels, increasing the demand for products and food and the concern about environmental crises, enzymes are seen as potential allies in the search for higher yield rates and lower environmental impacts [28]. In 2021 the global market of enzymes was availed as 6.4 billion dollars. This number will rise to 8.7 billion dollars by 2026, with a compound annual growth rate of 3.6% [29].

Many industries that use chemical transformation process have several disadvantages, both from a commercial and ecological point of view: non-specific reactions that lead to low yields, need to use conditions of elevated temperature and pressure that generate high energy costs and by-products with negative environmental impact, among others [30]. On the other hand, enzymatic reactions can, for the most part, be carried out under mild conditions of temperature and pressure and are highly specific, which generates a high yield rate and reduce chemicals use, water, energy, and waste generation, the reducing manufacturing impact on the environment [28,31,32].

Some advantages microbial enzymes show over other enzyme sources are easy handling and production, rapid multiplication under controlled conditions, easy genetic manipulation, high yield, greater stability and catalytic activity, greater economic viability, and regular supply due to the absence of seasonal fluctuations [28,33,34]. As a result, these enzymes have a wide range of applications in the most different industries, be it food, beverage, pharmaceutical, textile, toxic pollutants treatment and remediation, and biofuels, among others [33,35,36,37,38,39,40,41].

Some enzymes can degrade lignocellulosic biomass, showing potential for use in various sectors such as the food, textile, and biofuel industry [42,43,44,45]. So, in general, are hydrolases like the holocellulases (cellulases and xylanases) or oxidoreductases as laccase, manganese peroxidase, and lignin peroxidase collectively called ligninases [46].

## 3. Lignocellulolytic Enzymes

Lignocellulolytic enzymes constitute a group of enzymes capable of breaking lignocellulosic biomass into its various monomer units, making it accessible for bioconversion into multiple products and its applications in the most diverse industries [47,48,49]. Lignocellulosic biomass primarily consists of long cellulose and hemicellulose chains joined by lignin units and some non-structural soluble compounds [50]. Cellulose is the main component, constituting around 35% to 55% of lignocellulosic biomass, while hemicellulose, the second most abundant, makes up about 20% to 35% of the biomass. Lignin, the third principal component, constitutes 10% to 25% of lignocellulosic biomass [51].

At the molecular level, cellulose is characterized as a polymer formed by D-glucose units joined through β-1,4-glycosidic bonds [52,53,54]. According to the organizations’ degree of the bonds between the cellulose fiber chains, its structure alternates between a crystalline (monomers are organized, forming a more rigid structure) and a non-crystalline phase (monomers are not arranged) [55,56]. Unlike cellulose, hemicellulose has a highly branched and amorphous structure, with short side chains linked by β-1,4 glycosidic bonds, which confers less stability and degree of polymerization when compared to the first [57,58]. Generally, hemicellulose structure consists of multiple monomers of different monosaccharides, including pentose sugars (such as arabinose and xylose), hexose sugars (such as mannose, glucose, and galactose), uronic acids (such as methyl glucuronic, D-glucuronic, and D-galacturonic acids), in additions to other smaller groups [59,60]. According to its structure and the main sugar unit present in it, hemicellulose can be divided into four types of main structure: xylans, mannans, xyloglucans, and glucan [61]. Xylan, the main one, is a linear hemicellulose, and its main chain is composed of xylose units [62,63]. Finally, lignin consists of a complex, amorphous, and aromatic polymer, and acts as a glue, linking cellulose and hemicellulose molecules through covalent and hydrogen bonds. This structure forms a macromolecular complex that strengthens and gives greater rigidity and robustness to plant cell walls [64]. At the molecular level, lignin is characterized as an aromatic polymer composed of phenolic and non-phenolic parts, formed by three monolignol subunits: p-coumaroyl, coniferyl, and sinapyl alcohol [65,66,67]. Among them, a vast number of additional components can be present in lignin structure [65].

This way, these units forming lignocellulosic biomass are linked and organized in such a way as to form a molecular complex that has a highly resistant and recalcitrant structure [61,67,68,69]. Among several possible strategies for converting lignocellulosic biomass into fermentable sugars, the one that has been considered the most efficient and sustainable one is enzymatic hydrolysis. This process involves an intrinsic interaction of several enzymes, which can be obtained from different microorganisms [45,70,71]. This enzymatic system is composed of three types of main enzymes: cellulases, hemicellulases (hydrolytic enzymes that make cellulose and hemicellulose chains hydrolysis, respectively), and ligninases (oxidases and peroxidases which degrade lignin), besides some of the accessory enzymes [49].

### 3.1. Cellulases

Cellulases are the main enzymes capable of hydrolysis of β-1,4-glycosidic bonds present in cellulose molecules, releasing its monomeric subunits [72]. Taking into account the mode of action and substrate specificity, cellulases can be classified into three main types: (1) endoglucanases (EG); (2) exoglucanases (or cellobiohydrolases, CBHs); and (3) β-glycosidases (BG) [73]. Endoglucanases (EC 3.2.1.4) catalyze the random breakage of internal β-1,4-glycosidic bonds in amorphous regions of the cellulose chain, generating new ends that are exposed to the action of exoglucanases. Then, exoglucanases (EC 3.2.1.91; EC 3.2.1.176) cleave these reducing and non-reducing-ends in a processual way, releasing cellobiose. Finally, β-glycosidases (EC 3.2.1.21) are responsible for the last step in the hydrolysis of cellulose. These cleave and hydrolyze the previously generated cellobiose, releasing two glucose molecules. [74]. Thus, in order to occur complete and efficient cellulose hydrolysis, an elevated level of synergism between these enzymes is necessary [75]. 

Microorganisms, in general, have two ways of secreting cellulolytic systems [76]. Most aerobic cellulolytic microorganisms secrete cellulases as a set of individual enzymes, which act synergistically to break down cellulose [70]. Cellulases secreted this way contain a carbohydrate-binding module attached to the catalytic site by a flexible linker [77]. On the other hand, anaerobic cellulolytic microorganisms secrete cellulases in the form of multienzyme complexes with more than 1 million molecular weights, called cellulosomes [78,79]. In this complex, enzymes generally do not have a cellulose-binding module but are bound to accessory proteins which bind directly to cellulose [80]. The cellulosome complex was first discovered from the thermophilic anaerobic bacteria *Clostridium thermocellum* in the early 1980s. Since then, many cellulosome-producing anaerobic microorganisms have been identified and isolated from different ecosystems [81,82,83,84,85,86]. However, since the cellulosome complex makes it challenging to extract cellulases, aerobic microorganisms are usually preferred over anaerobic microorganisms for the industrial production of cellulases [76]. Some bacteria also have a third intermediate strategy for cellulase secretion, in which multifunctional enzymes containing two or more catalytic domains joined by a carbohydrate-binding module can be secreted either as free enzymes or incorporated into a cellulosic complex [87,88].

### 3.2. Hemicellulases

As with cellulose, the complete hydrolysis and degradation of hemicellulose also require the synergistic action of a set of enzymes [49]. These enzymes, called hemicellulases, act by breaking the existing glycosidic bonds between carbohydrates and carbohydrates, as well as supporting other glycohydrolases in removing methyl and acetyl groups on the hemicellulose surface [89]. Hemicellulases can be divided into two major classes: those with depolymerizing action, which hydrolyze the main chain glycosidic bonds (xylanases, glucanases, and mannanases), and accessory enzymes, which break the ester bonds and glycosidic bonds of hemicellulose side chains (α-L-arabinofuranosidase, acetyl xylan esterase, β-glucuronidase, glucuronyl esterase, and ferulic acid esterase) [90,91].

Among all, xylanases constitute the main class of enzymes that act in hemicellulose hydrolysis [49]. These, in turn, have the hydrolytic action of xylan, the principal constituent of hemicellulose, converting it into xylose and xylooligosaccharides such as xylobiose. Based on the mechanism of action and binding to substrate, xylanases can also be classified into different enzymes that constitute its enzymatic system, namely: endo-β-1,4-xylanases, β-D-xylosidases, α-glucuronidases, arabinases, acetyl xylan esterases, ferulic acid esterases and p-coumaric acid esterase [42]. Endo-β-1,4-xylanases (EC 3.2.1.8), the main components of xylanases, are those with endoxylanase activity, which break β-1,4-glycosidic bonds within the xylan chain, producing xylooligosaccharides and xylose units [60,92]. β-D-xylosidases (EC 3.2.1.37) act on these xylooligosaccharides non-reducing ends generated from xylan, successively removing their D-xylose residues [60,93,94]. α-glucuronidases (EC 3.2.1.139) act to cleave α-1,2-glycosidic bonds in glucuronic acid side chains of non-reducing units of xylose. At the same time, the arabinases (EC 3.2.1.55 and EC 3.2.1.99) are responsible for removing L-arabinose residues in xylose side chains [95,96]. Acetyl xylan esterases (EC 3.1.1.6), in turn, remove O-acetyl groups from acetyl xylan residues, while ferulic acid esterases (EC 3.1.1.-) and p-coumaric acid esterases (EC 3.1.1.-) cleave ester bonds in xylan, between the arabinose and ferulic acid side groups, and between arabinose and p-coumaric acid, respectively [42]. 

Several studies show that xylanases can be found in various sources, including bacteria, fungi, yeasts, algae, seeds, snails, and crustaceans. However, fungi and bacteria are recognized as the primary producers of these enzymes [34,42,97,98,99,100]. Based on their structure and the amino acid sequence, xylanases are mainly classified between 10 and 11 glycohydrolases families. Family 10 comprises high molecular weight enzymes composed of a cellulose-binding domain and another catalytic domain; these two are linked by a peptide. Therefore, this family mainly represents bacterial xylanases.

On the other hand, family 11, mostly belonging to fungi, is characterized by low molecular weight xylanases [101]. Furthermore, as discussed for cellulases, microorganisms have two ways of secreting xylanases. While aerobic fungi and bacteria do it as a set of individual enzymes, anaerobic fungi and bacteria secrete these enzymes in a cellulosome-like complex form called xylanosomes [102].

### 3.3. Ligninases

Since lignin consists of an aromatic and hydrophobic polymer composed of phenolic and non-phenolic parts, it exhibits a structural complexity that results in high resistance to hydrolase actions [68,103]. This way, its depolymerization, and degradation occur from several oxidative reactions, leading to the release of by-products with less stability. The enzymes involved in this enzymatic system of oxidative lignin degradation are called ligninases and can be of two main groups: peroxidases and oxidases [104,105].

Peroxidases are enzymes that initiate lignin depolymerization through oxidation reactions that result in free radicals and anions formation in the presence of hydrogen peroxide [49]. Among the peroxidases, four classes are known to act in this process: lignin peroxidases, manganese peroxidases, versatile peroxidases, and bleaching peroxidases [47]. Lignin peroxidases (EC 1.11.1.14) are glycoproteins containing heme groups, being central enzymes in lignin depolymerization. They perform the oxidation and degradation of various phenolic compounds by eliminating an electron in the presence of hydrogen peroxide as a substrate and may present different isoforms [106]. Manganese peroxidases (EC 1.11.1.13) are also heme-containing glycoproteins that oxidize a variety of phenolic and non-phenolic compounds in the presence of Mn and hydrogen peroxide as oxidizing agents [39]. These enzymes act on the oxidation of Mn^2+^ to Mn^3+^, which in turn oxidizes benzyl alcohol rings, thus causing lignin degradation [46]. Versatile peroxidases (EC 1.11.1.16) are enzymes that catalyze the oxidation of heterogeneous aromatic compounds using hydrogen peroxide as an electron acceptor. These enzymes integrate the oxidative properties of both lignin peroxidases and manganese peroxidases, oxidizing several phenolic and non-phenolic compounds in the presence of Mn^2^ [107]. Finally, the bleaching peroxidases (EC 1.11.1.19) constitute a new family of peroxidases containing a heme group. They are the main unit of the lignin degradation system in bacteria, capable of acting in a broad specificity of substrates and extreme pH [103].

The group of oxidases is mainly represented by laccases (EC 1.10.3.2), copper-containing enzymes that consist of monomeric, dimeric, and tetrameric glycoproteins [108]. These enzymes have three copper atoms linked coordinately to maintain the amino acid active sites [109]. In general, the oxidative activity of these enzymes occurs from the oxidation of four electrons of different aromatic and non-aromatic units of their substrates, followed by molecular oxygen reduction in the presence of water. These enzymes act on phenolic units’ oxidation, with an electron loss and consequent formation of unstable free radicals [110]. This reaction can be catalyzed by laccases either by direct or indirect substrate oxidation. At first, the substrate is oxidized due to direct contact with the enzyme’s copper. Second, substrate oxidation occurs through mediators in a two-step reaction: the first catalyzes the mediator, and then the catalyzed mediator oxidizes the substrate [46].

### 3.4. Accessory Proteins and Enzymes

In addition to the enzyme classes discussed above, the complete breakdown and degradation of lignocellulosic biomass also require the interaction of lignocellulolytic enzymes with other proteins, called accessory proteins [56]. The significant importance of this protein group is that they are directly involved in reducing biomass crystalline structure and recalcitrance, making the lignocellulose structure more susceptible to lignocellulase attack [111]. Therefore, accessory proteins assist in lignocellulolytic activity, either by breaking the hydrogen bonds in cellulose fiber or by oxidative mechanisms resulting in glycosidic bond breakdown [112]. Some accessory proteins already described include expansins and swolenins, which act in lignocellulolytic structure loosening and swelling, respectively, facilitating the access and activity of other enzymes [56,113,114]. In addition to these, some enzymes are also known to support lignocellulolytic activity. Examples of accessory enzymes are the LPMO (*Lytic Polysaccharide Monooxygenases*—EC 1.14.99.54). These enzymes, in turn, cleave β-1,4-glycosidic bonds of crystalline substrates such as cellulose and chitin, leading to the oxidation of C1 and C4 carbons [115]. This process then causes cellulosic fibers disorganization, facilitating cellulase access [56,112]. Among the LPMOs, the Auxiliary Activity Enzyme (AA9) is the most added to commercial enzyme cocktails [56]

## 4. Lignocellulolytic Microorganisms

Lignocellulolytic enzymes have already been reported from many microorganisms, being aerobic or anaerobic and living in the most diverse environments. This range of lignocellulolytic microorganisms includes fungi, bacteria, and archaea [77,116]. 

### 4.1. Archaea and Eubacteria

Some archaea have already been described as potential degraders of lignocellulosic biomass. Examples are species of the genera *Pyrococcus, Sulfolobus, Thermogladius,* and *Thermofilum* [116,117,118,119,120]. Most of these species live in extreme environments with elevated temperatures, pH, and salinity conditions. Due to their high thermoactivity and thermostability, these archaea are potential candidates for industrial processes that require extreme conditions, including pre-treatment and plant biomass conversion [116].

Due to their ability to adapt to pH and temperature changes, greater flexibility to oxygen demand, and potential use in genetic engineering, bacteria are also important sources for lignocellulase production [121,122,123,124], which have already been reported from several species of bacteria, including aerobic and anaerobic ones. However, they show significant differences in these enzyme production systems, yield rates, and final products of biomass degradation reactions [123]. The vast majority of these bacteria are reported from *Bacillus, Acinetobacter, Cellulomonas*, *Clostridium,* and *Pseudomonas*, although several other genera have already described species with lignocellulolytic potential [123,124]. 

Among the aerobics, actinobacteria stand out, which include species such as *Cellulomonas flavigena, Cellulomonas fimi, Actinomycosis bovis, Xylanimonas cellulosilytica* and *Thermobifida fusca* [125,126]. The latter contains both cellulolytic and lignocellulolytic enzymes, allowing its use for both cellulose hydrolysis and lignin modification [127]. Other bacteria studied with the potential for lignin breakdown are *Bacillus, Streptomyces, Sphingomonas, Pseudomonas, Rhodococcus,* and *Nocardia* [128]. Among anaerobic bacteria, those that stand out as lignocellulolytic enzyme producers belong to the *Clostridium* genre, such as *Clostridium thermocellum* [127]. Table 1 provides examples of bacteria producing different classes of lignocellulolytic enzymes.

### 4.2. Fungi

Although bacteria and archaea have some advantages in lignocellulase production, filamentous fungi are the most extensively studied microorganisms for lignocellulosic biomass breakdown [142,143,144]. This is due mainly to these organisms’ ability to secrete large amounts of enzymes in the extracellular environment, facilitating their obtaining process of them [70,145]. These fungi are formed by hyphae containing perforated walls called septa, which allow the passage of proteins and their secretion through the plasma membrane [146,147]. Among these, species of *Aspergillus* and *Trichoderma*, belonging to the phylum Ascomycota, are the best-known and applied examples of cellulolytic fungi in industry, accounting for more than 50% of studies related to cellulases [111,148,149,150,151]. These are known as soft rot fungi and cause cavities and erosions in plant cell walls [152]. Proteomic studies of *Aspergillus niger* and *Trichoderma reesei* showed that these fungi have an extensive secretome involved in lignocellulase degradation, containing different families of cellobiohydrolases, endoglucanases, β-glycosides and several hemicellulases [153]. In addition to these, other filamentous fungi genera such as *Penicillium, Fusarium,* and *Rhizopus* are also among the major industrial producers of these enzymes [111,154,155,156]. 

In addition to Ascomycetes, genomic analyzes of fungi belonging to Basidiomycota phylum showed that they have both enzymatic systems: a hydrolytic one for cellulose and hemicellulose degradation, and an oxidative one for lignin oxidation and degradation, the latter containing laccases and several peroxidases [157]. These basidiomycetes include white rot fungi and brown rot fungi, so called because they attack the plant cell wall leaving it with a fibrous texture in a bleached and brown color, respectively [157,158]. Among these, stand out species of some genera such as *Phlebia, Pleurotus, Phanerochaete, Trametes, Polyporus, and Lentinus*, among others [159,160]. Table 2 provides examples of distinct species of fungi capable of producing lignocellulolytic enzymes, described according to the class of enzyme produced.

### 4.3. Natural Habitat from Lignocellulolytic Microorganisms

Since microorganisms can exist in diverse habitats, the versatility of the lignocellulases they produce is also enormous [71]. Since the soil has immense microbial diversity, it is considered the most exploited environment for obtaining such enzymes [122,178]. In these environments, aerobic fungi primarily carry out lignocellulose decomposition [179]. However, in deeper soil layers, where the oxygen supply is limited, bacteria are the main degraders of lignocellulose [180]. In this same study by Wilhelm et al. was shown that, in forest environments, fungi are the ones with the most cellulolytic activity. At the same time, gram-negative bacteria are the microorganisms most involved in lignin decomposition. Soil microbial communities have been isolated and characterized with potential lignocellulolytic use [181,182]. 

Among the soil fungi known for plant biomass degradation are the genera *Trichoderma*, *Penicillium*, *Aspergillus*, *Humicola*, and *Fusarium* [45,71,143]. In a recent study by Shinde et al. [183], among different microorganisms isolated from soil (including fungi and bacteria), the fungal species belonging to *Trichoderma* and *Aspergillus* were those that demonstrated the highest enzymatic activity of lignocellulases, based on quantitative and enzymatic analysis. Several bacteria with lignocellulolytic capacity have also been isolated from the soil. The vast majority of studies have been carried out on species of *Bacillus*, *Pseudomonas*, *Serratia*, *Clostridium*, *Cellulomonas,* and *Streptomyces* [45,71,122]. 

Lignocellulolytic microorganisms from aquatic environments also have significant importance. Those from marine environments are the most sought-after due to the high industrial demand for stable enzymes under different conditions [71]. In addition, these microorganisms live in extremes of pressure, temperature, salinity, and diverse geochemical conditions. Thus, compared to terrestrial sources, enzymes derived from marine microbial sources are considered more potent for lignocellulosic biomass conversion [184]. Among these, *Bacillus* is one of the most reported as such, although several others have already been described ([45,71,184,185,186]. A study performed with different marine microorganisms and various lignocellulosic biomass sources showed that those with the highest lignocellulolytic activity were *Bacillus pumilus*, *Mesorhizzobium* spp., and *Aspergillus niger* and *Trichoderma viride* [187]. In addition, other endophytic fungi and bacteria are also seen as essential biomass degraders and lignocellulolytic enzyme producers [188,189,190,191].

Some microorganisms can also live in extreme temperatures, pressure, pH, saline, acidic or alkaline environments, among other conditions [192,193]. Because they also contain such properties and have excellent stability under extreme conditions, enzymes derived from these microorganisms are considered important biocatalysts for numerous biotechnological processes. Hence, they have gained significant interest recently [194,195,196,197]. These extremophilic microorganisms are also classified into several sub-groups [193]. Lignocellulases derived from psychrophilic microorganisms (able to adapt to very low temperatures, ranging from 15 °C to −40 °C) have antifreeze capacity and can maintain their catalytic activity even at temperatures below 0 °C [192,197]. Several psychrophilic fungi and actinobacteria already isolated showed lignocellulolytic enzyme production [45,198,199,200]. However, among the other psychrophilic bacteria, those capable of producing tolerant lignocellulases at low temperatures are restricted to a few species, such as *Pseudoalteromonas haloplanktis* and *Flavobacterium* spp., for example [201,202]. 

In addition to these, several thermophilic fungi (able to grow and develop at elevated temperatures, between 50 °C to over 100 °C) have been documented as efficient cellulase producers, such as *Aspergillus* spp., *Myceliophthora thermophila*, *Chaetomium thermophile, Humicola insolens,* and *Humicola grisea*, with activities ranging from 60–65 °C [203,204,205,206]. Bacteria also have a wide variety of thermophilic species described as lignocellulase producers, such as *Caldicellulosiruptor* spp., *Bacillus licheniformis*, and *Acidothermus cellulolyticus* [207,208,209]. Among them, someone’s stand out for being hyperthermophilic, growing around 100 °C and producing lignocellulases that maintain maximum activity up to temperatures around 80–106 °C, such as *Thermotoga* spp, for example [210,211].

Some microorganisms living in environments with extreme pH can also produce lignocellulases. In addition to maintaining their activity in extreme pH, many of these microorganisms (and their enzymes) also have other attributes, such as tolerance to saline environments, and thermophilic or psychrophilic capacity, among others. Distinct species of *Bacillus*, for example, have been reported to produce lignocellulases at pH ranging from 4 to 6.5 and optimal temperatures around 55–70 °C [212,213]. Furthermore, *Bacillus* spp. are also considered excellent producers of these enzymes at alkaline pH [214]. *Paenibacillus* spp. has also been described with lignocellulolytic activity at pH 4.0–5.5 and both low (20 °C) and high (50–70 °C) temperatures [215]. Several fungi can produce lignocellulases at acidic pH and elevated temperatures [45]. *Aspergillus* spp. isolated from various sources, for example, showed reasonable lignocellulases production rates at pH 2–4 and temperature at 50–80 °C [216,217,218,219]. On the other hand, few fungal species are known to produce such enzymes at alkaline pH, with most being restricted to *Paenibacillus* and *Aspergillus* [220]. 

The considerable advantage of pH and thermostable lignocellulase-producing microorganisms is that they make it possible to reduce or even eliminate heat and/or chemical pre-treatment steps, often necessary for lignocellulosic biomass conversion [71]. In addition, they present other additional advantages, such as ease of mixing, better substrate solubility, low risk of contamination, excellent storage stability, resistance to chemical denaturants and organic solvents, and increased reaction rates and catalytic activity, which makes them the most sought after in lignocellulosic industry [196,221,222].

## 5. Lignocellulolytic Enzymes Production

### 5.1. Methods for Enzymatic Production

The development of an enzymatic system for efficient lignocellulosic biomass hydrolysis on an industrial scale has been studied since the 1950s. Currently, two techniques are used for lignocellulase production, both on a laboratory and industrial scale: submerged fermentation (SmF) and solid-state fermentation (SSF). The first one can be defined as fermentation in the presence of excess water and consists of submerging the substrate in liquid, which requires the presence of free-floating liquid. On the other hand, solid-state fermentation is carried out on a solid and insoluble substrate without free liquid. In this type of fermentation, in addition to physical support, the substrate also provides the source of nutrients for the microorganism’s growth [223,224].

Since it allows greater control of parameters such as pH, temperature and agitation, easy recovery, and reproducibility, submerged fermentation is more used compared to solid-state fermentation for industrial production [224]. However, this technique also has disadvantages, such as high energy demand, the need for greater investment, and lower rates of productivity and yield in a longer fermentation time [225]. On the other hand, despite requiring a medium that maintains the necessary moisture for microbial growth, the solid-state fermentation technique presents a more accessible microorganism adaptation to the substrate, less contamination risk, and less water and energy needed, in addition to higher yield compared to submerged fermentation [226]. Thus, solid-state fermentation has been seen as an essential alternative for lignocellulosic biomass hydrolysis and subsequent production of by-products [223,227]. 

Different types of bioreactors can be used in both cases. In this way, bioreactors act as mechanical devices that provide in vitro conditions for microorganisms’ cultivation to obtain the desired products from the substrate [228]. For submerged fermentation, there are three types of bioreactors. The most used of them is the stirred tank bioreactor, which allows greater temperature control and oxygen transfer during the reaction. In addition, there is also the air transport bioreactor, which allows efficient mixing employing continuous fluid circulation through channels in a closed circuit, and the bubble column bioreactor, the most suitable for microorganisms sensitive to the carrier. For solid-state fermentation, the bioreactors used can be divided into four categories [224]. The bed is almost static in tray bioreactors, and air circulates freely around it. The bed remains static in a packed bed bioreactor, but the air is heavily left inside it. The bed is continuously stirred in a fluidized bed bioreactor, with air being strongly blown into it. Finally, in a swing drum bioreactor, the bed is constantly agitated in a drum so that air circulates freely above it without being forced to flow around [229].

Choosing an appropriate method for each case depends on numerous factors, including the microorganism and the substrate used. Overall, the main criteria for such a choice include: adequate mass transfer, low shear stress, sterility, aeration, pH and temperature control, low energy consumption, adequate material size, and an efficient mixing system [230]. For example, Pinheiro et al. [231] evaluated the laccase production from *Trametes versicolor* in three diverse types of bioreactors: stirred tank bioreactor, aluminum tray, and Erlenmeyer flasks. In this study, the authors found that the highest enzyme production rate occurred using stirred tank bioreactor, which may be related to the fact that it allows a greater oxygen supply to the microorganism. 

### 5.2. Types of Biomasses Used for Enzymatic Production

Regarding the substrate, diverse types can be used as carbon sources for lignocellulolytic enzyme production, including seeds, fruits, and agro-industrial residues [229,231,232,233,234,235,236,237]. Since it has low cost and wide availability, the use of agro-industrial residues contributes to reducing the production cost of these enzymes, in addition to not competing with the production of food for human and animal consumption [238]. In addition, it promotes a circular economy and more sustainable production, helping to reduce the impact on the environment of waste that would previously be largely disposed of incorrectly [239]. Some examples of agro-industrial waste used are: straw and rice bran, corn straw, coffee husks, sawdust, sugarcane bagasse, and waste from the paper industry, among others [240,241,242,243,244,245] (Figure 1).

Since complete biomass degradation requires a complex synergism between the several enzymes presented above, and their efficiency depends directly on the substrate and conditions used for this, there is a growing recognition that the use of a single type of enzyme produced from a single microorganism, is not the ideal approach for efficient biomass processing [246]. Thus, using a consortium of enzymes in the form of enzymatic cocktails seems to be the best strategy for a complete and more economical lignocellulosic biomass degradation. Furthermore, to allow the combined action of a pool of enzymes with different specificities, allowing one enzyme to act on the other product, in an enzyme cocktail, it is also possible to replace an individual enzyme in order to optimize each cocktail to a specific substrate and conditions [247]. 

In order to obtain an efficient cocktail, some parameters must be taken into account, such as microorganisms’ behavior when cultivated on different substrates, identifying the types of enzymes they produce in each case, and which enzymes are necessary for every kind of substrate degradation [60]. Various approaches can be used for enzymatic cocktails production, such as combinations of distinct species of fungi [248,249,250,251], bacteria [252,253,254], or even fungi and bacteria [255,256,257]. Furthermore, strategies for optimizing an enzymatic cocktail may include supplementation with a specific enzyme, be it hydrolytic or not, or supplementation with surfactants or other chemicals [247]. 

## 6. Application of Lignocellulolytic Enzymes

Lignocellulolytic enzymes have applications in the broadest sectors and industries, including: the food and beverage industry, pulp and paper industry, textile industry, biofuel production, and bioremediation (Figure 2). These enzymes can be applied in different processes and for various purposes.

### 6.1. Food and Beverage Industry

Lignocellulolytic enzymes have broad applications in the food and beverage industry. Along with pectinases, cellulases and xylanases are known as food-macerating enzymes [224]. These are applied in numerous processes, including extraction and clarification of fruit and vegetable juices, increasing yield, volatile characteristics, aroma, decreasing viscosity, and improving its property and the performance of the process as a whole [258,259]. Applying these enzymes for such a process is preferable to other conventional methods as they allow higher yields in less processing time and reduce thermal damage to the product [36]. In addition, lignocellulolytic enzymes can also be used to improve stability and texture, decrease the viscosity of fruit nectars and purees, and be important in the olive oil extraction process [260]. Among these enzymes, xylanases have the practical potential for use in the bakery industry. By hydrolyzing the hemicellulose of wheat flour, xylanases make the dough softer and delay crumb formation, allowing the dough to increase in volume and improve its quality [98]. Laccases also have considerable application in bread making, increasing stability and strength, enhancing softness, and decreasing read dough viscosity [261]. 

In addition, the animal feed also plays an essential role in the food industry as a whole, as it enables the production and distribution of animal protein [262]. However, the feed used in monogastric animals (swine and poultry) and ruminant rearing is usually composed of ingredients rich in lignocellulose, indigestible by these animals’ endogenous enzymes [263]. Thus, adding exogenous enzymes to this diet has been seen as an alternative to improve fiber degradability, increasing absorption efficiency, energy intake, and nutritional quality [264]. The most widely used lignocellulolytic enzymes for this purpose are cellulases, such as β-glucanases, and hemicellulases, such as mannanases and xylanases [264].

In the beverage industry, these enzymes are used to improve malt extraction efficiency for beer production, increasing fermentation rate and yield, as well as improving malt quality [224,259]. In this process, the use of xylanases also reduces beer’s muddy appearance and viscosity [36]. Laccases, in turn, have been used to remove unwanted phenolic compounds which cause browning and cloudiness and, in this way, improve the beers and other foods and beverages color [265]. Moreover, they can be used for the oxidation of polyphenols present in beer, contributing to the increased shelf life of both beers and wines. In wine production, the set of lignocellulolytic enzymes improves its coloring, clarification, and filtration, as well as quality and stability [266,267,268].

### 6.2. Textile Industry

In many textile industries, desizing, washing, and bleaching processes have used chemicals such as caustic soda, urea, acids, bases, and bleaches over the years. However, such products are toxic and cause environmental pollution when incorrectly disposed of [269]. Thus, enzymes are seen as a cleaner and more sustainable alternative for such processes [270]. In this scenario, the main lignocellulolytic enzymes’ application in the textile industry is through cellulase use in the biostoning process of cotton products. Furthermore, these enzymes can also be used in the washing process in order to selectively remove pectins, waxes, fats, minerals, natural dyes, and other impurities from cotton fabric [271].

The textile industry also uses cotton fiber bleaching processes to discolor its natural pigmentation so that it can later be dyed according to demand [272]. In addition to causing less damage to the fibers, the use of enzymes in this process also considerably saves the amount of water needed to do so. Moreover, laccases have a significant advantage since they act specifically on indigo dyes [273]. Laccases applied in the bleaching process have already been produced from different microorganisms, such as the fungi *Cerrena unicolor* and *Madurella mycetomatis*, and the bacterium *Brevibacillus agri*, which presented optimal temperatures ranging from 30 to 60 °C, and pH range from 3 to 6 [274,275,276]. Moreover, lignocellulolytic enzymes are also used in the polishing process, the last finishing step to improve fabric quality, in which fibers are hydrolyzed, providing a smoother surface with a clean, soft, and shiny touch. For example, Bussler et al. [277] applied cellulases produced from *Caulobacter crescentus* to jeans fibers, which, when analyzed by scanning electron micrographs, have shown a clean and smooth surface, indicating this enzyme potential for application in biopolishing of jeans. 

Finally, waste generated in the textile industry can also be treated using lignocellulolytic enzymes, mainly laccases, lignin peroxidases, and manganese peroxidases [278,279]. Unuofin [280] reported these dye’s discoloration using laccases produced from *Achromobacter xylosoxidans* and *Citrobacter freundii*, these having greater thermostability, with an optimal temperature ranging from 50 to 90 °C. This same author also has demonstrated the successful bleaching of synthetic dyes and jeans using laccases produced from *Pseudomonas* spp. These, in turn, had 80% of their residual activity recovered after the process’ extreme conditions, demonstrating significant tolerance to temperature, pH, salts, cations, and surfactants [280].

### 6.3. Pulp and Paper Industry

Aiming a more sustainable industrial production, the supply of virgin pulp for paper production has dropped significantly over the last few years. Because of this, paper industries have started to rely more on agricultural waste and paper waste [281]. To turn this waste into paper, mills incorporate many different processes, which include preparation, pulping, recovery, and bleaching [282]. Since the 1980s, the use of lignocellulolytic enzymes in this industry has increased significantly. These, in turn, have applications in numerous processes, such as pulp biobleaching and deinking, improved drainage, and effluent treatment. In addition, the enzymatic treatment also enhances the bleached pulp’s physical appearance, quality, and brightness. At the same time, ligninolytic enzymes can be used to treat toxic agents and other chemicals used in these processes [282,283,284,285,286]. Furthermore, enzymatic hydrolysis in the pulp and paper industry allows for less energy use and high selectivity, producing fewer harmful effects [287].

In the deinking process, these enzymes act to hydrolyze the bond between paper fibrils and ink particles, which are then removed using a flotation technique [288]. Biobleaching is a process in which lignin is separated from pulp to produce glossy white paper [285]. In this latter, xylanases are widely used as they attack hemicellulose, facilitating lignin release from cellulose [289,290]. In addition, cellulases are also especially important in reducing bleaching energy costs, increasing drainage efficiency, and improving paper gloss [283]. Furthermore, lignin oxidation by the action of laccases also significantly increases the final product brightness [291]. Several enzyme cocktails have been studied and produced for industrial pulp and paper processes [285]. Different cocktails of xylanases, laccases, and other enzymes produced from *Bacillus firmus*, *Bacillus pumilus*, *Bacillus nealsonii*, and *Bacillus halodurans* have demonstrated reasonable rates of kappa number reduction and chemical treatment, as well as increased pulp brightness [287,292,293,294]. Enzyme cocktails produced from *Aspergillus* spp. also proved to be very efficient in bioblanching, reducing considerably harmful agent use [295]. 

### 6.4. Biofuels Production

Biofuel production has grown significantly worldwide and is seen as one of the main alternatives to convert the planet’s climate crisis and greenhouse gas emissions [12]. As they do not compete with food crops, second-generation biofuels are seen as a more sustainable and efficient alternative when compared to first-generation ones [296]. These are produced from inedible lignocellulosic biomass, including agricultural and food processing residues [297,298]. Due to the lignocellulosic biomass recalcitrant structure, the sugars present in it are not fermented by first-generation bioethanol-producing microorganisms [299]. Thus, the production of second-generation biofuels is a process that involves pre-treatment processes (physical-chemical or biological), hydrolysis (acidic or enzymatic), and fermentation [300]. The recalcitrant lignocellulosic structure is disrupted in the pre-treatment stage, making it accessible to enzymatic activity [301]. Then, xylose and lignin are separated from cellulose, and cellulosic biomass undergoes enzymatic hydrolysis, transforming it into fermentable sugars. After hydrolysis, the sugars formed are fermented to produce biofuels, which can be biogas, bioethanol, or biohydrogen [302].

As they reduce the negative impacts on the environment, the use of lignocellulolytic enzymes in pre-treatment and hydrolysis stages has increased significantly [303]. Efficient enzyme cocktails for biofuel production include cellulases, hemicellulases, and ligninases, as well as pectinases and accessory proteins such as AA9 and swolenins [56]. In order to increase the feasibility of using microorganisms in biofuel production, recent research has sought to develop high-yield microorganisms for plant biomass degradation. In this scenario, genetic engineering of lignocellulosic biomass is one of the main strategies to increase biofuels [300,304,305]. On the other hand, genetic engineering and metabolic modulation of lignocellulolytic microorganisms is also an important strategy to improve both their enzymatic production and microorganism’s tolerance to inhibitors produced during the pretreatment step and to elevated temperature required conditions [300,306,307,308,309,310]. In addition, the high cost of commercial enzymes is still a limiting factor for large-scale biomass bioconversion. Estimates show that the production cost of these enzymes can reach $10.14/kg [311]. Thus, enzyme immobilization is an alternative that results in enzymatic hyperactivation and allows their various reuses [56,312,313]. Damásio et al. [312], for example, reported greater hydrolytic efficiency of arabinoxylan from co-immobilization of endo-xylanase and α-L-arabinofuranosidase from *Aspergillus nidulans* on glyoxyl agarose. 

### 6.5. Bioremediation

The industrialization and massive use of pesticides in agriculture are responsible for substantial amounts of residues and pollutants, contaminating soil, water, and air. In this sense, decontamination of these environments is one of the current major environmental challenges [314]. Bioremediation is a process that uses plants, microorganisms, or their enzymes to detoxify contaminants in soil, water, and other environments. This process may also include these contaminants’ partial or total transformation [314,315]. Several studies have revealed that oxidoreductase enzymes, such as the ligninases lignin peroxidase, manganese peroxidase, versatile peroxidase, and laccase, have biocatalytic activity with potential application for environmental pollutants degradation and mitigation [46,110,261,316]. Laccases, for example, catalyze oxidation-reduction reactions responsible for the biodegradation of several toxic substances, such as: phenolic compounds, pesticides, herbicides and fungicides, and pharmaceutical compounds, among others [265,317,318,319]. 

In this aspect, manganese peroxidases, lignin peroxidases, versatile peroxidases, and laccases produced by basidiomycete fungi are the most used to remove organic pollutants [320]. Among these, *Trametes* species are probably the most investigated and have already been commercialized by several companies [316,319]. Enzymes produced from *Trametes versicolor*, for example, showed enormous potential in the degradation of several types of pesticides [317,321,322], hospital waste [323], and pharmaceutical compounds [324], among others [320]. In addition to this, species such as *Pleurotus ostreatus, Phanerochaete chrysosporium,* and *Ganoderma lucidum* also demonstrate excellent enzymatic activities for different pollutants bioremediation [325,326,327,328,329,330]. However, current commercial prices of such enzymes are still too high for mass environmental applications [319]. Thus, developing novel approaches for genetic engineering, such as microorganisms and enzymes, may allow greater applications for degrading toxic compounds [331].

## 7. Recent Advances

### 7.1. Mixed Cultures

Mixed cultures consist of the growth of two or more microorganisms together under the same conditions [332]. This technique provides several benefits, such as better substrate utilization, greater adaptability to environmental changes, higher yield and productivity, and reduced contamination chances [333]. This can also be applied to the production of lignocellulolytic enzymes in order to have better conditions for obtaining enzyme cocktails. In addition to enabling the production of the complete set of cellulases, hemicellulases, and ligninases, mixed cultures also increase microorganisms’ growth rate and these enzyme levels of production compared to monoculture. Furthermore, the mixed cultures technique can be applied both under submerged and solid-state fermentation conditions [334].

In order to obtain an efficient mixed culture, the synergy between microorganisms is a crucial parameter. These must have similar optimal growth temperature, pH, and nutritional requirements [335]. Mixed cultures can be obtained from bacteria, fungi, and even from bacteria and fungi together [333,336,337]. Singh et al. [337], for example, have obtained higher bioethanol concentration and production from rice straw using a mixed culture of thermophilic anaerobic bacteria. However, the mixed cultivation from fungi often proves to be more accessible when compared to that from bacteria since, in their natural habitat, fungi grow symbiotically more easily on different substrates [334]. The optimization of lignocellulolytic enzyme production has already been obtained from different mixed cultures of *T. reesei* and *A. niger* using different substrates and conditions [335,338,339,340], in addition to *T. reesei* with other fungi, such as with *A. fumigatus* in sugarcane bagasse [341], with *Monascus purpureus* in wheat straw [342], and with *Penicillium citrinum* in wheat bran [334]. Silva et al. [343] also obtained greater hydrolysis efficiency of sugarcane bagasse with an enzymatic cocktail produced from the mixed culture of *T. reesei, A. brasiliensis*, *A. fumigatus*, and *Talaromyces* spp., with temperature and optimum pH of 50 °C and 4, respectively.

Furthermore, mixed culture between fungi and bacteria also presents an exciting strategy. Karuppiah et al. [344], for example, have obtained better conversion rates of several lignocellulolytic substrates from mixed cultures of *T. asperellum* and *B. amyloliquefaciens*. Furthermore, Preda et al. [336] have tested mixed cultivation of *Ganoderma lucidum* with 9 strains of bacteria for different ligninase production, of which lignin peroxidase was the one with the highest production increase rate. Finally, using a mixed culture of *B. licheniformis* and *S. cerevisiae*, Sharma et al. [345] obtained simultaneous saccharification and fermentation of wheat straw for bioethanol production, in addition to its significant increase. In this case, cellulases produced by the bacteria hydrolyzed wheat straw, while the yeast converted the sugar produced into ethanol.

### 7.2. Genetic Engineering

In addition to improving microorganisms’ cultivation in their natural form, another strategy that has been widely used to increase and enable the production and industrial-scale application of lignocellulolytic enzymes is the genetic engineering [346]. In this sense, several techniques can be used, such as directed evolution, gene editing, and heterologous expression. Furthermore, there are many possible aspects to be improved, including the catalytic activity, stability, and resistance to inhibitors, as well as microorganisms’ regulatory networks, metabolism, and morphology and increasing the synergism and efficiency of enzyme cocktails [346]. For example, to improve the catalytic efficiency of cellulases produced by *T. reesei*, Jiang and collaborators [347] replaced the native gene cbh I of this microorganism with its counterpart from *Chaetomium thermophilum*, which led to an increase in cellulase activity in 2.2-fold.

It is also possible to build chimeric complexes with enzymes from different microorganisms through gene editing. Brunecky et al. [348], for example, have produced a chimeric cellulase containing an endoglucanase and cellulose-binding domains from bacteria and a cellobiohydrolases domain from fungi. This, in turn, showed high enzyme activity when compared to uncomplexed cellulases. Improving lignocellulolytic enzyme stability under adverse environmental conditions is also highly sought, as it allows hydrolysis at elevated temperatures, which is often necessary for industrial biomass conversion processes [349]. Using targeted evolution, a recent study has demonstrated 820-fold increased thermostability in the GH11 family [350]. Furthermore, by introducing disulfide bonds into the xylanase structure of *T. reesei*, Tang et al. [351] have demonstrated their greater acid and alkaline resistance. In addition to these aspects, genetic engineering also makes it possible to increase enzyme resistance and tolerance to different inhibitors [352].

However, these enzymes will only have industrial applications when they can be produced at high productivity rates at a low cost. Therefore, another preference that has been studied is the engineering of lignocellulolytic microorganisms. Sequence mutations in transcription factors, heterologous expression, and the development of CRISPR/Cas-9-based genome editing methods are some techniques that have driven advances in this area [352]. The transcription of lignocellulolytic enzymes is induced by specific inducers and repressed by repressor molecules. Thus, this enzyme expression is controlled by a network of regulatory mechanisms mediated by multiple transcription factors [353,354,355,356,357]. This way, modifying the regulatory network of these microorganisms’ transcription factors for expressing the lignocellulolytic enzyme is a vital strategy [358]. In *T. reesei*, for example, the transcription factor CRE1 is the main repressor of lignocellulolytic enzyme expression, while XYR1 is its main transcriptional activator. Mutations that caused the first silencing and the second’s overexpression demonstrated elevated levels of extracellular cellulase secretion, producing hypersecretory *T. reesei* strains [359,360]. The combinatorial engineering of three transcriptional activators in *P. oxalicum* (ClrB, XlnR, and AraR) also demonstrated the generation of a strain with an increase in lignocellulolytic enzymes production from 3.1 to 51.0-fold, in addition to a more significant release of fermentable sugars from corn fiber, when compared with the original strain enzymes [361].

Another focus of lignocellulolytic microorganism genetic engineering is to modify their metabolic network [352]. Some examples of what has been done in this regard include eliminating specific proteases to decrease the degradation of the lignocellulolytic enzyme [362], improving strain growth and protein production rate [363], and metabolic control in order to balance other enzymes synthesis that compete for precursors and energy [364,365], and the alteration of fungal mycelia morphology in order to decrease the medium viscosity, allowing greater mass transfer and oxygen supply [366,367]. Concerning the heterologous expression of lignocellulolytic enzymes, it is sought to express a functional lignocellulolytic system in order to allow non-lignocellulolytic microorganisms to hydrolyze and transform lignocellulosic biomass. The most commonly used non-lignocellulolytic microorganisms for this purpose are *Zymomonas mobilis*, *Escherichia coli*, *Pichia pastoris*, and *Saccharomyces cerevisiae* [277,368,369,370,371]. The production of recombinant lignocellulases may be the solution to limitations of high substrate cost and maintenance of the necessary conditions for these enzymes production, as well as more resistant and stable strains production and higher rates of enzyme production [372,373,374,375]. However, these modified enzymes and microorganisms still lack broad industrial application, so efforts must be made to optimize these aspects [352].

### 7.3. Bioprospecting

Native microorganisms can be found in the most diverse environments and produce several enzymes with industrial importance activities. Therefore, bioprospecting for new microorganisms and lignocellulolytic enzymes is a valuable tool that has been increasingly researched and used [376]. Bioprospecting involves screening native strains and enzymes from various sources (soil, water, air) for specific traits based on high yields of desired end products [376,377]. One approach that has been widely used is to look for specific genomic content in environmental samples through metagenomics [378,379]. To isolate new genes and pathways encoding enzymes or biosynthesis of biomolecules, functional metagenomics has been widely successful in isolating and identifying new families of proteins, especially lignocellulolytic enzymes [380]. Using metagenomics, therefore, allows the prospection of potential lignocellulolytic microorganisms very quickly, in addition to allowing the identification of both cultured and non-cultured microorganisms [381].

Several studies have already shown the efficiency of bioprospecting in searching for lignocellulolytic microorganisms aiming at producing enzymatic cocktails. To date, metagenomic analyses have also resulted in the identification of numerous potential lignocellulolytic enzymes [233,249,380,382,383,384,385,386]. Shotgun analysis of a bacterial consortium enriched with carboxymethylcellulose, for example, resulted in the reconstruction of six complete genomes, four of which were new, including *Bacillus thermozeamaize*, *Geobacillus thermoglucosidasiu*, and *Caldibacillus debillis*. CAZy analysis of these genomes revealed the presence of several genes associated with lignocellulosic material degradation and an abundance of GHs [387]. The search for and isolation of microorganisms in extreme environments that produce enzymes with such properties also concentrates a large part of bioprospecting efforts [388,389]. Bioprospecting of cellulolytic microorganisms from the Red Sea (an environment with elevated temperature, salinity, and low nutrients levels), for example, resulted in bacterial strains with high cellulase production, demonstrating that this environment can be an important source of these microorganisms [390]. Different bacteria isolated from mangrove soil also have shown potential hemicellulolytic capacity, indicating that these environments represent a promising source for enzyme bioprospecting due to their characteristics, such as fluctuations in oxic/anoxic and salinity conditions [391]. The search for lignocellulolytic microorganisms in cold environments also presents relevant results. Brück et al. [200] have shown that the search for filamentous fungi in Ecuadorian soil resulted in strains with high cellulase and xylanase activity at a temperature of 8 °C, demonstrating potential application in bioremediation processes and effluent treatment under cold weather conditions. Thus, bioprospecting allows the identification of new and potential microorganisms and lignocellulolytic enzymes in the most diverse environments.

## 8. Future Perspectives

The lignocellulolytic enzyme market is expanding and is projected to grow even more in the coming years. Despite being produced by numerous microorganisms and having a wide variety of applications, lignocellulolytic enzymes still have some barriers to wide industrial use. Among these barriers is mainly the production cost of these enzymes. In this sense, developing enzymatic cocktails is essential for cost reduction. Therefore, in-depth studies on the physiology and metabolism of these fungi are of significant importance to optimize their cultivation conditions, in addition to further studies to optimize appropriate proportions of each enzyme in the construction of a cocktail. Lignocellulolytic enzyme stability is also one factor that affects their applicability in several processes. Although several research are carried out to improve and change enzyme stability, efforts are still needed in the search for strategies and production conditions that favor enzyme stability, in addition to the search for new lignocellulolytic enzymes in extreme environments in which these enzymes have greater stability. The development and modification of lignocellulolytic microorganisms’ strains by different genetic engineering techniques have already brought many advances to optimization and increase of lignocellulolytic enzymes production rates, as well as to the reduction of this process cost. However, the techniques for GMOs (Genetically Modified Organisms) production and the vectors currently used for heterologous expression still have difficulties to be faced. Some examples are the need for methanol (a toxic agent) to induce expression in *P. pastoris*, the hyperglycosylation of proteins expressed in *S. cerevisiae*, and the formation of inclusion bodies of proteins expressed in *E. coli*. Thus, it is necessary to search for new microorganisms and vectors that increase and facilitate the heterologous expression of lignocellulolytic enzymes without requiring additional steps. More research is also needed in bioprospecting novel microorganisms and undiscovered enzymes with lignocellulolytic potential. After the discovery, technological advances will be required to help replicate the ideal environmental conditions for its growth in the laboratory and, later, on an industrial scale.

## 9. Conclusions

Lignocellulolytic enzymes are a vital alternative to change of chemical agents in the most diverse industries, such as the textile, pulp, paper, food and beverage, biofuels, and bioremediation, among others, contributing to making them more sustainable processes and mitigate the current environmental crisis effects. Enzymes’ global market is expanding, within which microorganisms, the most outstanding representatives of the planet’s biodiversity, are considered the main sources of these enzymes. Among these, filamentous fungi are the most researched and used, as they allow the production of substantial amounts of enzymes extracellularly, facilitating their obtainment. Because they inhabit the most diverse environments on the planet, lignocellulolytic microorganisms can also be found in extreme temperature, pH, and oxygen concentration conditions, producing lignocellulolytic enzymes that are more resistant and stable to industrial requirements. However, enzyme production on an industrial scale still presents many obstacles. In order to circumvent this situation, several research have been carried out in search of new strategies, including mixed cultivation, genetic engineering, and bioprospecting techniques. Thus, with the expansion of studies in search of greater viability of these techniques, new sources of lignocellulolytic enzymes, cost reduction, optimization of production conditions, and lignocellulolytic enzymes application, this market may prove to be increasingly promising.

## Figures and Tables

**Figure 1 microorganisms-11-00162-f001:**
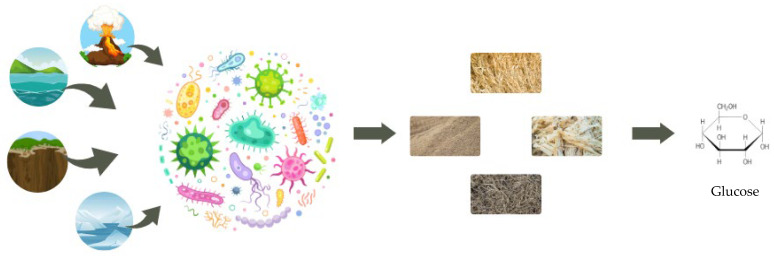
Microorganisms from the most diverse habitats are able, through the production of lignocellulolytic enzymes, to degrade various types of biomasses such as rice straw, corn straw, sugarcane bagasse, and soybean straw. From this, fermentable sugars are formed, exemplified here by glucose, which has a wide range of industrial applications.

**Figure 2 microorganisms-11-00162-f002:**
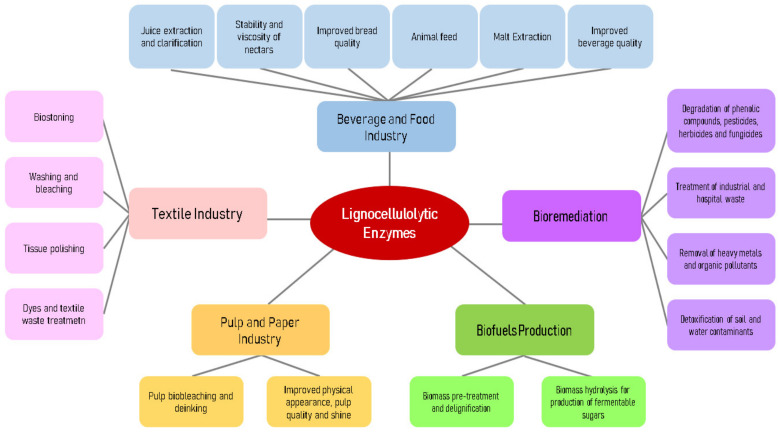
Different applications of lignocellulolytic enzymes.

**Table 1 microorganisms-11-00162-t001:** Examples of bacteria producing lignocellulolytic enzymes.

Enzymes	Microorganism	Optimal pH	Optimal Temperature	Reference
Endoglucanases	*Bacillus subtilis*	5	60	[129]
	*Neobacillus sedimentimangrovi*	7	60	[130]
	*Arthrobacter woluwensis*	8	50	[131]
	*Thermotoga naphtophila*	6	90	[132]
Exoglucanases	*Clostridium thermocellum*	5.7	70	[133]
Xylanases	*Thermotoga marítima* TmxB	5	100	[134]
	*Acinetobacter johnsonii*	6	55	[135]
	*Bacillus haynesii*	7	40	[136]
	*Caldicoprobacter algeriensis*	6.5	80	[137]
Peroxidases	*Pseudomonas* spp.	3–8	20–80	[138]
	*Bacillus ayderensis* SK3-4	7	75	[139]
Laccases	*Lysinibacillus macroides*	7	30	[140]
	*Pseudomonas parafulva*	8	50	[141]

**Table 2 microorganisms-11-00162-t002:** Examples of fungi producing lignocellulolytic enzymes.

Enzyme	Microorganism	Optimal pH	Optimal Temperature (°C)	Reference
Endoglucanases	*Trichoderma viride*	5	40	[161]
	*Cladosporium cladosporioides*	4	30	[162]
	*Fusarium* spp.	5.5	30	[163]
	*Aspergillus niger*	5.5	30	[164]
Exoglucanases	*Trichoderma viride*	5	40	[161]
	*Fusarium* spp.	5.5	30	[163]
	*Aspergillus niger*	5.5	30	[164]
	*Phaeolus spadiceus*	4.5	25–30	[165]
β-glycosidases	*Trichoderma viride*	5	40	[161]
	*Cladosporium cladosporioides*	4	30	[162]
	*Aspergillus niger*	5–9	25–45	[166]
	*Fusarium* spp.	5.5	30	[163]
Xylanases	*Trichoderma* spp.	5	28	[167]
	*Trichoderma harzianum*	6	70	[168]
	*Aspergillus tubingensis*	3–8	30–60	[169]
	*Talaromyces amestolkiae*	7	30	[170]
Peroxidases	*Pleurotus ostreatus*	3.3	25	[171]
	*Hypsizygus ulmarius*	7	28	[172]
	*Pleurostuus florida*	7	28	[172]
	*Phlebia radiata*	3	80	[173]
Laccases	*Trametes polyzona*	4.5	55	[174]
	*Trametes versicolor*	4–5	40–50	[175]
	*Coriolopsis gallica*	6–8	40–60	[176]
	*Pycnoporus* spp.	6	0	[177]

## Data Availability

Data sharing is not applicable to this review.

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
