# Peer review of "Lignocellulolytic Biocatalysts: The Main Players Involved in Multiple Biotechnological Processes for Biomass Valorization"

_microorganisms, 2023, doi:10.3390/microorganisms11010162_

Round 1

Reviewer 1 Report

In the review article entitled “Lignocellulolytic Microorganisms: the main representatives of the biodiversity involved in multiple biotechnological processes for biomasses degradation” by Ana Laura Totti Benatti et al., the authors report a quite extensive compendium about lignocellulolytic enzymes and microorganisms. As the authors state in the abstract: “this review proposes to discuss the most critical aspects of the enzymes forming lignocellulolytic complex and the main microorganisms used to obtain them”…” different possible industrial applications for these enzymes will be discussed, as well as information about their production modes and considerations about recent advances and future perspectives in research in pursuit of expanding lignocellulolytic enzyme uses at an industrial scale”.

However, I must point out that despite their clams in the abstract, this review (although well written and smoothly readable) does not represent a critical and in-depth analysis of the available literature. Still, I think the manuscript could deserve publication after revision.

Major points:

Although I have appreciated the authors’ intention to provide a compendium about all the enzymes discussed, I think there is a disproportion about the amount of text dedicated to the description of the single enzymes’ classes and the description of more relevant industrial aspects, i.e., in the section: “LIGNOCELLULOLYTIC ENZYMES PRODUCTION”. I can understand that for a naïve reader it could be useful to have much information about each enzyme and microorganism, but it would be more useful if the authors will focus the dissertation more about the opportunities and challenges of applying such enzymes at industrial level.

I will not go in details of the single paragraph, but as an example regarding “Pulp and Paper Industry “, the authors are addressed to the following paper, as an example of industrial application of such enzymes:

Almeida, Nazaré, et al. "Use of a Novel Extremophilic Xylanase for an Environmentally Friendly Industrial Bleaching of Kraft Pulps." International Journal of Molecular Sciences 23.21 (2022): 13423.

As it concerns the paragraph “Biofuels Production”, the review would improve if some more information would be provided about the different industrial settings and how the enzymes could help to tackle specific pitfalls. The authors are addressed to the following paper, as an example about the description of the different industrial setting that are possible in this field:

Zuliani, Luca, et al. "Biorefinery gets hot: thermophilic enzymes and microorganisms for second-generation bioethanol production." Processes 9.9 (2021): 1583.

Concerning the authors’ statement “different possible industrial applications for these enzymes will be discussed, as well as information about their production modes and considerations about recent advances and future perspectives in research in pursuit of expanding lignocellulolytic enzyme uses at an industrial scale”…I miss much about the production modes of the enzymes they describe.

Minor points:

What do the authors mean with “the enzymes forming lignocellulolytic complex”? This definition seems to limit the description only to cellulosome-forming enzymes.

I would suggest editing the title:

From :“Lignocellulolytic Microorganisms: the main representatives of the biodiversity involved in multiple biotechnological processes for biomasses degradation” 

to: “Lignocellulolytic Biocatalysts: the main players involved in multiple biotechnological processes for biomass valorisation".

Author Response

Ribeirão Preto, December 12, 2022.

To:

Ms. Ruth Lin

Assistant Editor of MicroorganismsParte superior do formulário

Manuscript ID: Microorganisms-2063073

We are submitting a new revised version of the review #Microorganisms-2063073 entitled now as "Lignocellulolytic Biocatalysts: the main players involved in multiple biotechnological processes for biomass valorization”. We appreciate the attention of the reviewer and all the suggestions received to improve our research. The answers to all questions are listed below. In addition, we have revised the entire text and highlighted the changes in yellow. We await a favorable response to the publication of our review.

I look forward to hearing from you.

Dr. Maria de Lourdes T. M. Polizeli

Response to Reviewer 1

1) In the review article entitled “Lignocellulolytic Microorganisms: the main representatives of the biodiversity involved in multiple biotechnological processes for biomasses degradation” by Ana Laura Totti Benatti et al., the authors report a quite extensive compendium about lignocellulolytic enzymes and microorganisms. As the authors state in the abstract: “this review proposes to discuss the most critical aspects of the enzymes forming lignocellulolytic complex and the main microorganisms used to obtain them”…” different possible industrial applications for these enzymes will be discussed, as well as information about their production modes and considerations about recent advances and future perspectives in research in pursuit of expanding lignocellulolytic enzyme uses at an industrial scale”.

However, I must point out that despite their clams in the abstract, this review (although well written and smoothly readable) does not represent a critical and in-depth analysis of the available literature. Still, I think the manuscript could deserve publication after revision.

 A: Thank you for your comment and point of view on the article. We aimed to describe a state-of-the-art update of different aspects of microorganisms and their lignocellulolytic enzymes. In this way, the material presented was relatively large. A more critical and detailed review of the issues addressed would make the review even more extensive, which we did not intend. Therefore, we have removed the word “critical” from the abstract for clarity.

2) Although I have appreciated the authors’ intention to provide a compendium about all the enzymes discussed, I think there is a disproportion about the amount of text dedicated to the description of the single enzymes’ classes and the description of more relevant industrial aspects, i.e., in the section: “LIGNOCELLULOLYTIC ENZYMES PRODUCTION”. I can understand that for a naïve reader it could be useful to have much information about each enzyme and microorganism, but it would be more useful if the authors will focus the dissertation more about the opportunities and challenges of applying such enzymes at industrial level.  I will not go in details of the single paragraph, but as an example regarding “Pulp and Paper Industry “, the authors are addressed to the following paper, as an example of industrial application of such enzymes:

Almeida, Nazaré, et al. "Use of a Novel Extremophilic Xylanase for an Environmentally Friendly Industrial Bleaching of Kraft Pulps." International Journal of Molecular Sciences 23.21 (2022): 13423.

 As it concerns the paragraph “Biofuels Production”, the review would improve if some more information would be provided about the different industrial settings and how the enzymes could help to tackle specific pitfalls. The authors are addressed to the following paper, as an example about the description of the different industrial setting that are possible in this field:

Zuliani, Luca, et al. "Biorefinery gets hot: thermophilic enzymes and microorganisms for second-generation bioethanol production." Processes 9.9 (2021): 1583.

A: Thank you very much for your comment and suggestion, which we fully agree with. However, the article does not bring more in-depth aspects of the industrial applications of lignocellulolytic enzymes. On the contrary, this article intends to address microorganisms and their lignocellulolytic enzymes, providing information about their possible applications so that the reader can seek more information in the cited literature. We believe the current review will contribute to many scholars on the subject.

3) Concerning the authors’ statement “different possible industrial applications for these enzymes will be discussed, as well as information about their production modes and considerations about recent advances and future perspectives in research in pursuit of expanding lignocellulolytic enzyme uses at an industrial scale”…I miss much about the production modes of the enzymes they describe.

A: Thank you very much for your observation. We agree that the modes of production were not widely covered. Therefore, additional information on this subject has been added on page10 so that the reader can improve his knowledge and seek more information in the cited literature.

4) What do the authors mean with “the enzymes forming lignocellulolytic complex”? This definition seems to limit the description only to cellulosome-forming enzymes.

A: We appreciate your comment and observation and fully agree with them. The term "lignocellulolytic complex" may limit its description to cellulosome-forming enzymes, so this term has been replaced by "lignocellulolytic system".

5) I would suggest editing the title: From :“Lignocellulolytic Microorganisms: the main representatives of the biodiversity involved in multiple biotechnological processes for biomasses degradation”  to: “Lignocellulolytic Biocatalysts: the main players involved in multiple biotechnological processes for biomass valorisation".

A: Thank you very much for your comment and suggestion. The title has been changed as suggested. The new title will value more the current revision.

Reviewer 2 Report

The manuscript „ Lignocellulolytic Microorganisms: the main representatives of the biodiversity involved in multiple biotechnological processes for biomasses degradation” presents the state of the art and a very well discussion aspects of the main microorganisms used to obtain lignocellulose degradable enzymes and their usage. The structure of the article and the presentation of the literature is very good. In the enzymatic use, the cost of the enzymes plays a crucial role. The author also addresses this in two passages (Page 15 “current commercial prices of such enzymes are still too high…” Page 18 “barriers is mainly the production cost”). In addition, the authors could also indicate the costs of the enzyme production and the enzyme use. All in all, a very nice literature review that adds value to existing literature reviews for the reader.

Author Response

Ribeirão Preto, December 12, 2022.

To:

Ms. Ruth Lin

Assistant Editor of MicroorganismsParte superior do formulário

Manuscript ID: Microorganisms-2063073

We are submitting a new revised version of the review #Microorganisms-2063073 entitled now as "Lignocellulolytic Biocatalysts: the main players involved in multiple biotechnological processes for biomass valorization”. We appreciate the attention of the reviewer and all the suggestions received to improve our research. The answers to all questions are listed below. In addition, we have revised the entire text and highlighted the changes in yellow. We await a favorable response to the publication of our review.

I look forward to hearing from you.

Dr. Maria de Lourdes T. M. Polizeli

Response to Reviewer 2

1) The manuscript  “Lignocellulolytic Microorganisms: the main representatives of the biodiversity involved in multiple biotechnological processes for biomasses degradation” presents the state of the art and a very well discussion aspects of the main microorganisms used to obtain lignocellulose degradable enzymes and their usage. The structure of the article and the presentation of the literature is very good. In the enzymatic use, the cost of the enzymes plays a crucial role. The author also addresses this in two passages (Page 15 “current commercial prices of such enzymes are still too high…” Page 18 “barriers is mainly the production cost”). In addition, the authors could also indicate the costs of the enzyme production and the enzyme use. All in all, a very nice literature review that adds value to existing literature reviews for the reader.

A: Thank you very much for the positive comments about the article and the observation made. We agreed that the cost of enzymes could also be indicated, and this was added on page 15 in “Biofuels Production”’.
